# Learning Distinguishable Trajectory Representation with Contrastive Loss

**Tianxu Li**[1,2]    **Kun Zhu**[1,2,*]  **Juan Li**[1]   **Yang Zhang**[1]

[1]College of Computer Science and Technology, Nanjing University of Aeronautics and Astronautics, China
[2]Collaborative Innovation Center of Novel Software Technology and Industrialization
{tianxuli, zhukun, yangzhang, juanli}@nuaa.edu.cn

## Abstract

Policy network parameter sharing is a commonly used technique in advanced deep multi-agent reinforcement learning (MARL) algorithms to improve learning efficiency by reducing the number of policy parameters and sharing experiences among agents. Nevertheless, agents that share the policy parameters tend to learn similar behaviors. To encourage multi-agent diversity, prior works typically maximize the mutual information between trajectories and agent identities using variational inference. However, this category of methods easily leads to inefficient exploration due to limited trajectory visitations. To resolve this limitation, inspired by the learning of pre-trained models, in this paper, we propose a novel Contrastive Trajectory Representation (CTR) method based on learning distinguishable trajectory representations to encourage multi-agent diversity. Specifically, CTR maps the trajectory of an agent into a latent trajectory representation space by an encoder and an autoregressive model. To achieve the distinguishability among trajectory representations of different agents, we introduce contrastive learning to maximize the mutual information between the trajectory representations and learnable identity representations of different agents. We implement CTR on top of QMIX and evaluate its performance in various cooperative multi-agent tasks. The empirical results demonstrate that our proposed CTR yields significant performance improvement over the state-of-the-art methods.

## 1   Introduction

Cooperative multi-agent reinforcement learning (MARL) can provide effective collaboration among agents and has shown promise for solving real-world multi-agent tasks, such as robot swarms [Hüttenrauch et al., 2017], autonomous driving [Bhalla et al., 2020, Dinneweth et al., 2022], and wireless communications [Li et al., 2022b]. However, effective collaboration in complex multi-agent tasks still remains a challenge for MARL. One of the key issues is that the joint action-observation space grows exponentially in size with the number of agents, which highlights an urgent demand for the scalability of MARL algorithms.

To address the scalability issue, learning decentralized policies for agents has been widely adopted. This allows agents to make action decisions based on their partial observations. However, learning a private decentralized policy network for each agent in MARL may require training a large amount of policy network parameters, resulting in inefficient learning. To enhance learning efficiency, many advanced MARL algorithms adopt the parameter sharing technique, including policy gradients [Lowe et al., 2017, Ma et al., 2021, Wang et al., 2020d, Ndousse et al., 2021, Zhang et al., 2021] and

---

*Corresponding author.

38th Conference on Neural Information Processing Systems (NeurIPS 2024).

value-based algorithms [Iqbal et al., 2021, Yang et al., 2021, Wang et al., 2020a, Sunehag et al., 2018, Rashid et al., 2018]. Incorporating parameter sharing enables all agents to make action decisions using a shared policy network. This significantly reduces the number of policy network parameters. Additionally, training a shared policy network facilitates the sharing of experiences among all agents, alleviating the unstable learning problem arising from partial observability. These advantages of parameter sharing dramatically improve the learning efficiency and accelerate the training speed of MARL algorithms [Wang et al., 2020b].

However, although parameter sharing has many advantages, agents sharing the policy network parameters tend to become homogeneous since they typically learn similar behaviors under similar observations [Hu et al., 2022, Mahajan et al., 2019], resulting in inefficient exploration and poor diversity. Challenging multi-agent tasks typically require extensive exploration and diverse policies among agents. For example, in a football game, agents in a team require to play different roles such as goalkeeper, defender, midfielder, and forward, taking diverse tactics to achieve more credits. If they behave similarly to compete for a ball, they may not achieve satisfactory results.

One of the most common methods to encourage multi-agent diversity is to maximize the mutual information between trajectories and agent identities by using variational inference methods [Jiang and Lu, 2021, Li et al., 2021] that learn parameterized trajectory discriminators to distinguish the trajectories of different agents given agent identities. However, due to the high mutual dependence between agent identities and trajectories, the agents tend to frequently visit known trajectories that contain more identity information, where they can achieve larger rewards than discovering new trajectories, leading to serious overfitting of trajectories to agent identities. Consequently, despite the emergence of diversity among agents, the agents unfortunately suffer from inefficient exploration.

To encourage multi-agent diversity while guaranteeing efficient exploration, we propose a novel Contrastive Trajectory Representation (CTR) method based on learning distinguishable trajectory representations, which encourages multi-agent diversity in an abstract contrastive representation space. Our motivation is that, although the shared policy network may receive similar inputs, it can still learn diverse representations, leading to varied behaviors. Unlike previous mutual information-based methods using variational inference, our method adopts a novel contrastive learning lower bound for the mutual information between trajectory representations and learnable identity representations. Notably, the learnable identity representation introduced in our method differs entirely from the fixed agent identity used in prior works. It is trained by minimizing the contrastive learning loss in order to constrain the trajectory representations of different agents to be linearly classified. As a result, the distinguishability among trajectory representations can be achieved and does not depend on any fixed agent identity. The learned distinguishable trajectory representations can then be used in the downstream action-decision tasks to learn more diverse and exploratory policies.

Our contributions can be summarized as follows: first, we propose a novel method for encouraging multi-agent diversity through learning distinguishable trajectory representations, which minimizes the contrastive learning loss between trajectory representations and identity representations of different agents. The distinguishable trajectory representations do not rely on fixed agent identities and thus lead to more efficient exploration; second, to reduce the gap between the contrastive learning lower bound and the mutual information objective caused by the small size of the dataset storing trajectory representations, we further extend the contrastive learning loss by increasing the number of negative samples; third, we provide a practical learning framework for CTR and apply our approach to QMIX; forth, we evaluate CTR in both grid world environments and the StarCraft Multi-Agent Challenge (SMAC) benchmark. The empirical results demonstrate that CTR significantly outperforms the existing state-of-the-art methods and yields more exploratory and diverse policies.

## 2 Backgrounds

### 2.1 Multi-Agent System

We consider learning in the fully cooperative multi-agent Decentralized Partially Observable Markov Decision Process (Dec-POMDP) [Oliehoek and Amato, 2015] described as a tuple $\langle A, S, U, P, R, O, \Omega, \gamma \rangle$, where $A$ represents a set of $|A|$ agents, $s \in S$ is the global state of the environment, and $U$ is a set of agents' actions. At the beginning of each time step, each agent $a$ receives an observation $o^a \in \Omega$ according to the function $O(s, a)$ and then selects an action $u^a \in U$. All the agents' actions compose a joint action $\boldsymbol{u}$, and the environment then transitions to the next

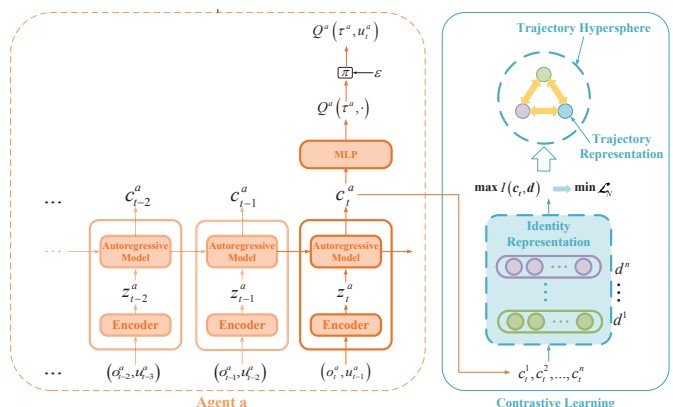

Figure 1: Architecture of CTR model.

state $s'$ with the probability drawn from the transition function $P(s' \mid s, \boldsymbol{u})$. At the same time, the environment feeds back to the agents a shared team reward $r = R(s, \boldsymbol{u})$. $\gamma \in [0, 1)$ is a reward discount factor. The observation-action pairs $\langle o^a, u^a \rangle$ of agent $a$ make up its trajectory $\tau^a \in \mathcal{T}$. Each agent $a$ learns its individual policy $\pi^a(u^a \mid \tau^a)$, forming a joint policy $\boldsymbol{\pi}$, to maximize the joint action-value function $Q^{\boldsymbol{\pi}}(s, \boldsymbol{u}) = \mathbb{E}_{s_{0:\infty}, \boldsymbol{u}_{0:\infty}} \left[ \sum_{t=0}^{\infty} \gamma^t r_t \mid s_0 = s, \boldsymbol{u}_0 = \boldsymbol{u}, \boldsymbol{\pi} \right]$.

## 3 Contrastive Trajectory Representation

### 3.1 Motivation and Intuitions

To promote multi-agent diversity, the agents need to learn diverse policies. In order to achieve this purpose, prior works [Jiang and Lu, 2021, Li et al., 2021] have devoted to maximizing the mutual information between trajectories and agent identities of different agents via variational inference methods. However, this category of methods forces the agent to visit known trajectories where they can achieve larger rewards than discovering new ones, making the learned policy prone to overfitting. The theoretical analysis of this limitation is provided in Appendix A. To address this issue, the intuition behind our proposed CTR is that we can instead learn diverse policies from the trajectory representations distributed on a contrastive representation hypersphere. In this section, we show how to learn trajectory representations and how the learned distinguishable trajectory representations can be used to learn diverse policies in practical learning algorithms.

### 3.2 Contrastive Trajectory Representation

In this section, we introduce the details of our CTR model that encourages multi-agent diversity by learning distinguishable trajectory representations.

The architecture of the CTR model is shown in Figure 1. First, a non-linear encoder $C_{enc}$ maps the input assembled by the observation of agent $a$ at time step $t$ $o_t^a$ as well as the last step action $u_{t-1}^a$ to a latent representation $z_t^a = C_{enc}(o_t^a, u_{t-1}^a)$. Next, to encode the previous action-observation sequences of the trajectory, an autoregressive model $C_{gar}$ is used to summarize all the latent representations $z_{\leq t}^a$ and generate a trajectory representation $c_t^a = C_{gar}(z_{\leq t}^a)$. The trajectory representation $c_t^a$ can empirically achieve better performance compared with the latent representation $z_t^a$ when used to make action decisions in the Dec-POMDP setting since it stores additional information from the historical trajectory that can alleviate the non-stationarity issue caused by the partially observable constraints [Sunehag et al., 2017]. In practice, for simplicity, we adopt standard network structures such as resnet blocks for the encoder and GRUs for the autoregressive model. It is notable that other types of encoder and autoregressive models can also be employed in the CTR model.

Next, we introduce how to train the CTR model to learn distinguishable trajectory representations, encouraging multi-agent diversity. It can be difficult to enlarge the distance between the trajectory representations of different agents directly. Instead, we can introduce an additional learnable identity representation for each agent to represent the agent identity. Unlike previous works [Jiang and Lu,

2021, Li et al., 2021] that use fixed one-hot vectors to represent the agent identities, in this paper, the identity representation adopted in our method is a learnable vector trained to linearly classify the trajectory representations of different agents.

Concretely, at the beginning of the training process, we randomly initialize a learnable vector $d^a \in \mathbb{R}^H$ for each agent $a$ as its identity representation that has the same dimensions as the trajectory representation. To achieve the distinguishability between trajectory representations of different agents, we maximize the mutual information between the trajectory representations and identity representations of agents:

$$I(c_t; d) = \mathcal{H}(c_t) - \mathcal{H}(c_t \mid d) = \mathbb{E}_{c_t, d} \left[ \log \frac{p(c_t \mid d)}{p(c_t)} \right], \tag{1}$$

where $\mathcal{H}$ is the entropy. Estimating the mutual information directly is typically intractable. In this paper, we present a novel method to solve the objective of mutual information between the trajectory representations and identity representations, unlike previous variational inference methods. Concretely, inspired by contrastive learning [Chen et al., 2020], a popular self-supervised learning method for learning representations, we use a contrastive learning loss, or the InfoNCE loss [Oord et al., 2018], to derive and optimize a tractable lower bound for the mutual information:

$$I(c_t; d) \geq \log(|A|) - \mathcal{L}_N, \tag{2}$$

where $|A|$ is the number of agents, and $\mathcal{L}_N$ is the InfoNCE loss. Note that $\log(|A|)$ is a constant. Therefore, by minimizing the $\mathcal{L}_N$, we maximize the mutual information $I(c_t; d)$. We next design a practical contrastive learning loss to learn distinguishable trajectory representations. Given a set of trajectory representations of all agents at time step $t$, $\mathcal{C} = \left\{ c_t^{a'} \right\}_{a'=1}^{|A|}$, and agent $a$'s identity representation $d^a$, the goal of contrastive learning is to make sure that the identity representation of agent $a$ $d^a$ is close with its corresponding trajectory representation $c_t^a$ while being distant from other trajectory representations in $\mathcal{C} \setminus \{c_t^a\}$. To achieve this goal, we minimize the contrastive learning loss:

$$\mathcal{L}_N = - \mathbb{E}_{(d^a, \mathcal{C}) \sim \mathcal{D}} \left[ \log \frac{f\left(c_t^a, d^a\right)}{\sum_{c_t^{a'} \in \mathcal{C}} f\left(c_t^{a'}, d^a\right)} \right] \tag{3}$$

where $f\left(c_t, d\right) = \exp\left(c_t^T d\right) \in \mathbb{R}$. $c_t^T d$ measures the similarity between the trajectory representation $c_t$ and the identity representation $d$. Minimizing the contrastive learning loss trains both the CTR model and the identity representations. Here, the identity representation of each agent serves as a linear classifier, linearly classifying the trajectory representations output by the CTR model for minimal contrastive learning loss. As a result, the distinguishability among trajectory representations can be achieved.

## 3.3   Multi-Agent Contrastive Learning Loss

One limitation of applying the contrastive learning loss given by Equation 3 to the multi-agent setting is that the small size of dataset $\mathcal{C}$, which is equal to the number of agents, induces a larger gap between the true mutual information objective and the contrastive learning lower bound, which can hurt the performance. The contrastive learning lower bound requires a larger number of samples to tighten its value to the true mutual information [Oord et al., 2018]. To resolve this problem, we consider extending the Equation 3 to the contrastive learning loss with $|A|$ positive samples:

$$\mathcal{L}_N^m = -\mathbb{E} \left[ \frac{1}{|A|} \sum_{a=1}^{|A|} \log \frac{|A| f\left(c_t^a, d^a\right)}{\sum_{a'=1}^{|A|} f\left(c_t^{a'}, d^{a'}\right) + \sum_{a'=1}^{|A|} \sum_{c_t^{a''} \in \mathcal{C}, a'' \neq a'} f\left(c_t^{a''}, d^{a'}\right)} \right]. \tag{4}$$

The contrastive leaning loss $\mathcal{L}_N^m$ shown in Equation 4 calculates the expectation over $|A|$ positive pairs $\{c_t^a, d^a\}_{a=1}^{|A|}$ and $|A|(|A| - 1)$ negative pairs $\{\{c_t^{a'}, d^a\}_{c_t^{a'} \in \mathcal{C}, a' \neq a}\}_{a=1}^{|A|}$. Notably, $\mathcal{L}_N^m$ actively

increases the number of negative samples in the denominator from $O(|\mathcal{C}|)$ to $O\left(|\mathcal{C}|^2\right)$ that can help the contrastive learning loss with a smaller dataset $\mathcal{C}$ to lead to more stable and robust results in challenging multi-agent tasks. By minimizing the $\mathcal{L}_N^m$, the trajectory representations of all agents stay close to their corresponding identity representations while being far away from other identity representations, leading to the distinguishability among trajectory representations. We refer the reader to Appendix C for the Pytorch-style pseudocode of our proposed CTR.

**Differences to contrastive learning** We note that the contrastive learning employed in our method is quite different from its common usage in self-supervised pre-training that learns representations by contrasting the positive and negative pairs of instances. In our method, we apply contrastive learning to learn representations in a fully supervised manner by introducing an identity representation for each agent. This allows the trajectory representations belonging to the same agent to be pulled together on the trajectory representation hypersphere, while simultaneously pushing apart trajectory representations from different agents. The fully supervised manner ensures that the minimization of the contrastive learning loss entails increased distances between trajectory representations of different agents.

### 3.4 Learning Algorithm

In this section, we discuss how to integrate CTR with existing MARL algorithms adopting the decentralized policy, to encourage multi-agent diversity. As illustrated in Figure 1, we implement CTR in individual policy networks of agents to learn distinguishable trajectory representations. The overall learning framework of CTR should consist of two parts: (i) the RL loss function of the MARL algorithm to train the decentralized policy towards maximizing the environment returns; (ii) the contrastive learning loss of CTR as an auxiliary loss function to train the decentralized policy in order to learn distinguishable trajectory representations. Thus, we can formulate the total loss function as follows:

$$\mathcal{L}_{total} = \mathcal{L}_{RL} + \alpha \mathcal{L}_N^m, \tag{5}$$

where $\mathcal{L}_{RL}$ is the RL loss function and $\mathcal{L}_N^m$ is the contrastive learning loss function. $\alpha$ is a hyperparameter adjusting the weight of contrastive learning loss $\mathcal{L}_N^m$ compared with the RL loss $\mathcal{L}_{RL}$. The overall CTR framework is trained end-to-end in a centralized manner. In this paper, we consider integrating CTR with the value-decomposition framework where each agent learns its policy through optimizing an approximation of the joint action-value function, denoted by $Q_{tot}(s, \boldsymbol{u})$, as follows:

$$\mathcal{L}_{TD}(\theta) = \sum_{i=1}^{b}\left[\left(r + \gamma \max_{\mathbf{u}'} Q_{tot}\left(s', \mathbf{u}'; \theta^-\right) - Q_{tot}(s, \mathbf{u}; \theta)\right)^2\right] \tag{6}$$

where $b$ is the batch size of transition samples, $\theta$ and $\theta^-$ represent the parameters of $Q_{tot}$ and target $Q_{tot}$, respectively, as in DQN [Mnih et al., 2013]. $Q_{tot}$ is a combination of per-agent utilities $Q_a$ where the decentralized policies are derived. We consider using QMIX [Rashid et al., 2018] to decompose the combination $Q_{tot}$, which factors the $Q_{tot}$ into a monotonic nonlinear combination of agent utilities. Thus, we can train the overall CTR framework end-to-end by minimizing:

$$\mathcal{L}_{total} = \mathcal{L}_{TD}(\theta) + \alpha \mathcal{L}_N^m. \tag{7}$$

The TD loss $\mathcal{L}_{TD}(\theta)$ trains both the mixing network and agent utility networks. The contrastive learning loss $\mathcal{L}_N^m$ trains both the identity representations and agent utility networks (including the autoregressive model as well as the encoder of CTR). It is notable that CTR is a component of the agent utility network to learn distinguishable trajectory representations via contrastive learning. Thus, CTR won't break the IGM rule [Rashid et al., 2018] of QMIX. Moreover, our method only adds one linear layer (identity representation), resulting in a small overhead that would not decrease the learning efficiency of the integrated learning algorithm. This is crucial for the parameter sharing mechanism since we do not need to train an additional neural network at the expense of learning efficiency (the advantage of parameter sharing) for promoting multi-agent diversity. We refer the reader to Appendix B for the implementation of CTR on top of the policy gradient method.

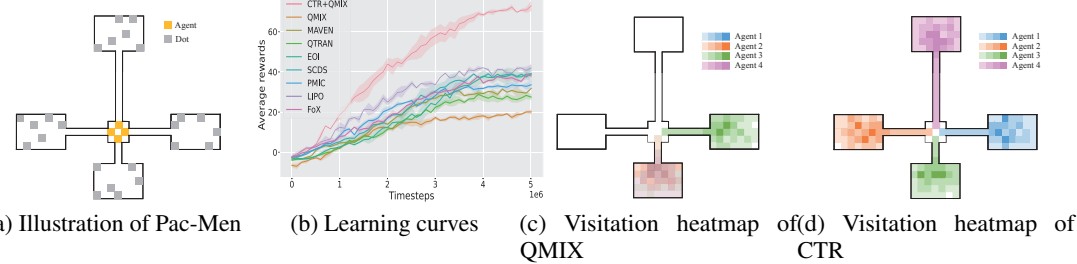

| (a) Illustration of Pac-Men | (b) Learning curves | (c) Visitation heatmap of QMIX | (d) Visitation heatmap of CTR |

Figure 2: The performance comparison between our proposed CTR and baselines in Pac-Men.

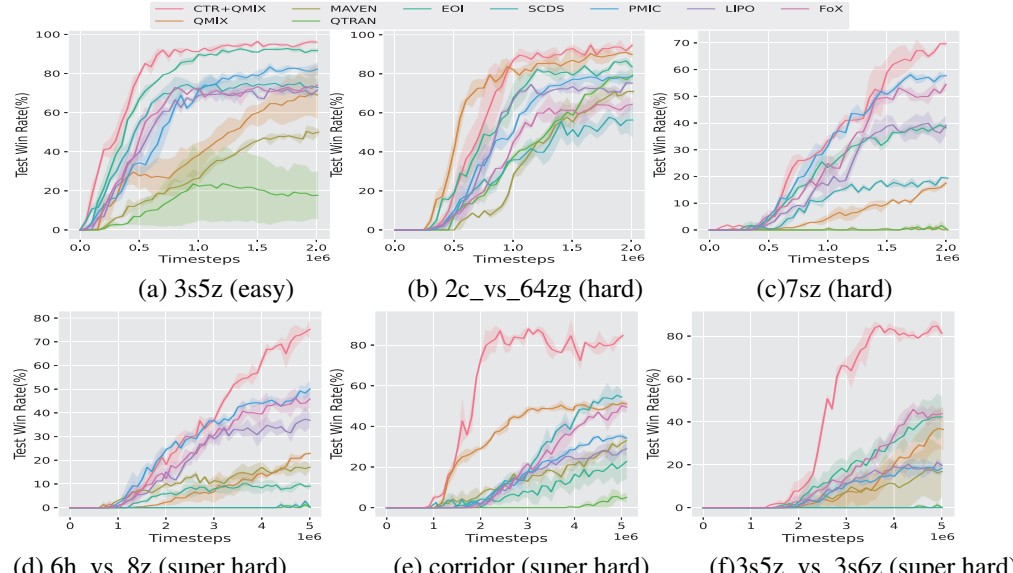

| (a) 3s5z (easy) | (b) 2c_vs_64zg (hard) | (c)7sz (hard) |
| (d) 6h_vs_8z (super hard) | (e) corridor (super hard) | (f)3s5z_vs_3s6z (super hard) |

Figure 3: Performance comparison between our proposed CTR and baselines in SMAC scenarios. Without loss of generality, all results are presented with the mean and standard deviation of performance tested with five random seeds.

## 4 Experiments

In this section, to demonstrate the outperformance of our proposed CTR, we evaluate CTR in Pac-Men, SMAC, and SMACv2 benchmarks. We compare our method with various state-of-the-art algorithms: value-decomposition algorithms (QMIX [Rashid et al., 2018], QTRAN [Son et al., 2019]) and mutual information-based exploration methods (MAVEN [Mahajan et al., 2019], EOI [Jiang and Lu, 2021], SCDS [Li et al., 2021], PMIC [Li et al., 2022a], LIPO [Charakorn et al., 2023], and FoX [Jo et al., 2024]). To ensure a fair comparison, we set the same values for common hyperparameters across different methods. The hyperparameters used in our experiments and training details are provided in Appendix G. Moreover, CTR does not include the agent identity in the input of the shared policy network to take action decisions.

### 4.1 Pac-Men

We first design a grid-world environment called Pac-Men, as shown in Figure 2a, to demonstrate the effectiveness of CTR in encouraging multi-agent diversity. In Pac-Men, four agents are initialized at the center of the maze. Each agent has a partial observation and can only observe a 4×4 grid around them. The goal of each agent is to eat the dots randomly initialized in edge rooms. We set different lengths for the paths towards edge rooms to improve the task difficulty and only the downward path is within the agent's observation scope, highlighting an urgent demand for efficient exploration. The

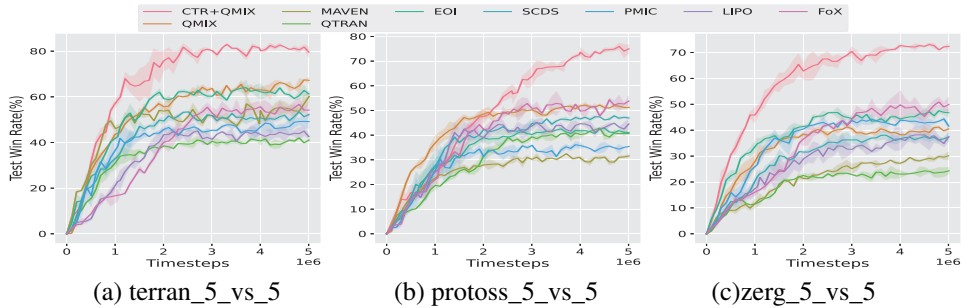

(a) terran_5_vs_5      (b) protoss_5_vs_5      (c)zerg_5_vs_5

Figure 4: Performance comparison between our proposed CTR and baselines in SMACv2 scenarios.

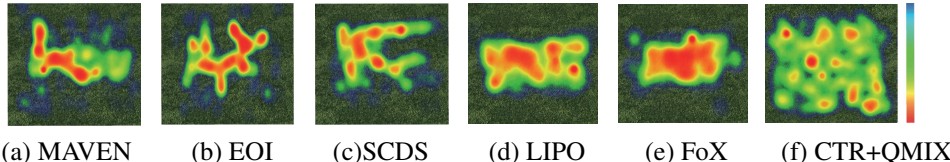

(a) MAVEN    (b) EOI    (c)SCDS    (d) LIPO    (e) FoX    (f) CTR+QMIX

Figure 5: Visitation heatmaps of different algorithms in the terran_5_vs_5 scenario.

performance of CTR and baselines is shown in Figure 2b. We note that QMIX falls into the local optimum and does not yield satisfactory performance in Pac-Men since some agents learned similar policies and went to the same edge room, resulting in ineffective competition for dots among agents. This can be verified by the visitation heatmap of QMIX shown in Figure 2c, where three agents go to the bottom edge room. In contrast, CTR achieves significantly superior to all baselines in Pac-Men, avoiding falling into local optimum through optimizing the objective of mutual information between the identity representations and trajectory representations using contrastive learning loss. As shown in Figure 2d, the diverse policies learned by CTR enable the agents to go to different edge rooms, leading to efficient cooperation among agents. The mutual information-based baselines achieve similar performance since they fail to find the edge room with the longest path due to inefficient exploration.

## 4.2 SMAC and SMACv2

The StarCraft Multi-Agent Challenge (SMAC) [Samvelyan et al., 2019] is a common-used benchmark for evaluating cooperative MARL algorithms. To demonstrate the effectiveness of our proposed CTR, we conduct experiments in 6 SMAC scenarios: 3s5z (easy), 2c_vs_64zg (hard), 7sz (hard), 6h_vs_8z (super hard), corridor (super hard), and 3s5z_vs_3s6z (super hard). Note that the performance is not comparable between different versions of SMAC. We use the SC2.4.10 version of SMAC.

The test win rates achieved by our method and baselines in different SMAC scenarios are shown in Figure 3, demonstrating the outperformance of our proposed CTR in the SMAC scenarios compared to baselines. In the super hard scenarios: 6h_vs_8z, corridor, and 3s5z_vs_3s6z, where enemies are more powerful than agents, CTR significantly outperforms all baselines. Qmix fails to learn optimal policies in these scenarios since these scenarios typically require agents to learn to distribute the enemies' attack due to a large strength gap between agents and enemies, which necessitates the emergence of diverse policies. CTR dramatically improves the final performance of QMIX by learning distinguishable trajectory representations. Compared to other mutual information-based baselines, CTR is more robust in promoting multi-agent diversity and achieves impressive final performance in super hard scenarios. EOI performs poorly because its probabilistic trajectory classifier overfits to the agent identities, hindering efficient exploration.

Moreover, homogeneous behaviors such as 'focus fire' are desired in the easy scenario 3s5z to quickly defeat enemies. Note that CTR would not hinder such homogeneous behaviors that can lead to more environmental rewards and conversely achieves satisfactory performance. We refer the reader to Appendix E for further evaluations of our method in the scenarios requiring homogeneous behaviors.

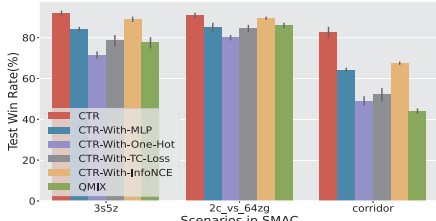

Figure 6: Performance comparison between CTR and ablation variants in SMAC scenarios.

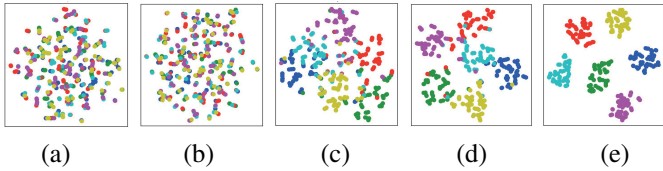

(a)       (b)       (c)       (d)       (e)

Figure 7: T-SNE plots of trajectory representations of different agents learned by different variants of CTR ((a) QMIX (b)CTR-With-One-Hot (c) CTR-With-TC-Loss (d) CTR-With-InfoNCE (e) CTR), emerging in the corridor scenario of SMAC.

**Stochasticity and Exploration** Due to the fixed team compositions and initial positions in the scenarios of SMAC, the agents can easily win the game when they master the particular action sequences, such as 'kiting', demonstrating that the SMAC scenarios lack efficient stochasticity to test the exploration of MARL algorithms. We further conduct experiments on a more challenging benchmark called SMACv2 [Ellis et al., 2022] that enables stochasticity in SMAC scenarios via introducing random team compositions and random start positions.

We conduct experiments in three SMACv2 scenarios: terran_5_vs_5, protoss_5_vs_5, and zerg_5_vs_5. Figure 4 shows the win rates for CTR and baselines. CTR yields more robust performance than baselines and successfully adapts to the stochasticity from environments. We attribute this to the efficient exploration strategies derived from distinguishable trajectory representations. The mutual information-based baselines do not achieve satisfactory performance in SMACv2 scenarios. These algorithms are prone to overfitting since they prefer to visit known trajectories that contain more identity information than exploring new trajectories, resulting in poor exploration. This can be verified by the visitation heatmaps provided in Figure 5, where the movements of agents trained by the mutual information-based methods are located in the fixed partial areas of the map. In contrast, CTR incentivizes more efficient exploration since CTR prevents overfitting by mapping the trajectories onto a contrastive representation hypersphere, where the representations only need to satisfy the distinguishability constraint. As a result, CTR yields more exploratory policies. The movements of CTR agents are uniformly distributed on the map.

### 4.3 Ablation Study

In this section, we conduct ablation studies on the proposed CTR framework to investigate the contributions of the main components in the CTR framework, including (A) autoregressive model, (B) identity representation, and (C) contrastive learning loss with $|A|$ positive samples. To test component A, we design CTR-With-MLP that ablates the autoregressive model employed in CTR by replacing it with multiple layer perceptions (MLPs). To test component B, we design CTR-With-One-Hot, which ablates the learnable identity representation by using a pre-defined one-hot vector to represent the identity of an agent. To test component C, we design two variants: CTR-With-TC-Loss and CTR-With-InfoNCE. CTR-With-TC-Loss optimizes a trajectory classification (TC) loss that directly predicts the corresponding agent identities given the trajectory representations regardless of negative samples. CTR-With-InfoNCE performs the vanilla contrastive learning loss given by Equation 3 for $|A|$ times to distinguish the trajectory representations of all agents.

We test these variants in three SMAC scenarios: 3s5z (easy), 2c_vs_64zg (hard), and corridor (super hard). The results are shown in Figure 6. CTR-With-One-Hot performs worst among all ablations, with similar performance to QMIX. This performance decline indicates that the pre-defined one-hot

vector fails to enable the agents to learn diverse behaviors. We further provide the t-SNE plots, as shown in Figure 7, for learned trajectory representations. It becomes apparent that in the case of CTR-With-One-Hot, the trajectory representations of different agents are mixed together as in QMIX. Similarly, we notice the performance decrease in CTR-With-TC-Loss. This degradation arises from the TC Loss's primary focus on agent identity prediction, essentially learning a maximum likelihood function over the collected trajectories given agent identities. However, it does not incorporate constraints aimed at ensuring the separation between trajectory representations of different agents. Consequently, trajectory representations of different agents learned by CTR-With-TC-Loss stay close to each other without a distinct gap as demonstrated in Figure 7, making it challenging to distinguish them efficiently. In contrast, CTR robustly constrains the trajectory representations of the same agent to be close and those of different agents to be far apart on the trajectory representation hypersphere.

CTR-With-InfoNCE consistently achieves better performance than CTR-With-TC-Loss across all scenarios, demonstrating the benefits brought by contrasting negative samples when learning distinguishable trajectory representations. However, CTR-With-InfoNCE results in obvious performance degradation in the super hard corridor scenario. This phenomenon indicates that the contrastive learning loss given by Equation 4 adopted in our CTR method that involves more negative samples when learning trajectory representations leads to more stable and robust results. As illustrated in Figure 7, the trajectory representations of different agents learned by CTR entail an increase in distances compared to those learned by CTR-With-InfoNCE. CTR-With-MLP does not lead to an obvious performance drop in the 3s5z (easy) and 2c_vs_64zg (hard) scenarios. However, in the super hard corridor scenario, CTR-With-MLP yields a dramatic decrease in performance, demonstrating that learning the context of the trajectory using an autoregressive model is beneficial for agents to improve their performance.

## 5   Related Works

The diversity in MARL encourages the differences between the policies of agents. SVO [McKee et al., 2020] studies multi-agent diversity by drawing on the social value orientation, a theory from social psychology, to solve multi-agent social dilemmas. It verifies the utility of population heterogeneity in cooperative MARL. SVO realizes the social value orientation via forming an intrinsic reward to encourage diverse policies. RODE [Wang et al., 2020c] achieves multi-agent diversity by assigning various actions to restricted roles. RODE is efficient when the agent has a small action space that can be decomposed. It can be inefficient for RODE to be applied in continuous action with huge action space.

MAVEN [Mahajan et al., 2019] proposes to enable the value-based agents to condition on a shared latent variable controlled by a hierarchical policy. To learn diverse joint behaviors, MAVEN maximizes the mutual information between the latent variable and the trajectories. Another method EOI [Jiang and Lu, 2021] proposes to encourage the individuality of agents by a supervised learning method that trains a probabilistic classifier to learn a probability distribution over agents with regard to their observations. CDS [Li et al., 2021] adopts the objective of mutual information to encourage multi-agent diversity. CDS optimizes the mutual information by formulating lower bounds derived by the Boltzmann softmax distribution and the variational inference, respectively. PMIC [Li et al., 2022a] maximizes the mutual information associated with superior cooperative behaviors while minimizing the mutual information related to inferior ones. CIA [Liu et al., 2023] realizes the credit-level distinguishability of value-decomposition based methods by identifying the temporal credits of different agents. LIPO [Charakorn et al., 2023] considers policy compatibility as a means to learn diverse behaviors, identifying the unique behaviors of each policy by optimizing the mutual information objective. FoX [Jo et al., 2024] presents formation-based exploration, which encourages the exploration of various formations by guiding agents to thoroughly comprehend their current formations. These works have shown promise in encouraging multi-agent diversity. However, they overemphasized learning the dependence between the agent identity and trajectory, forcing the agent to frequently visit similar observations, preventing the agents from further exploration.

## 6   Limitations and Future works

As our method needs to contrast all agents' trajectories, this necessitates that the training process should be centralized so that the trajectories of all agents can be collected. Thus, our method cannot

be applied to fully decentralized MARL methods. Moreover, in contrastive learning, by collecting many negative samples, the model is challenged to distinguish the positive pair from a larger pool of negatives. This helps the model learn more robust and discriminative features. Although we developed multi-agent contrastive learning loss to increase the number of negative samples, however, the number of negative samples is still limited if the total number of agents in multi-agent environments is very small. For our future work, we may develop efficient methods to augment existing trajectory samples to increase the number of samples.

Despite the emergence of multi-agent diversity, we also note the need for homogeneous behaviors. Although our method would not impede the learning of homogeneous behaviors that can lead to more environmental rewards, how to control diversity automatically can be an interesting direction for our future work.

## 7    Conclusion

In this paper, we consider learning distinguishable trajectory representations over raw trajectories to encourage multi-agent diversity. Our method achieves distinguishability among trajectory representations by maximizing the mutual information between trajectory representations and identity representations of agents through the minimization of the contrastive learning loss. We evaluate our method in different challenging cooperative tasks, and it demonstrates a significant performance improvement over existing state-of-the-art methods. Our simple yet effective method reveals the importance of representation learning in promoting efficient exploration, leading to optimal policies.

## Acknowledgments and Disclosure of Funding

This work was supported in part by National Natural Science Foundation of China (62061146002), and in part by Natural Science Foundation of Jiangsu Province (Grant No. BK20211567, BK20222012).

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

# A   Theoretical analysis of limitations of existing methods

This section analyzes the limitation of existing methods that the agents tend to visit known trajectories rather than exploring new ones from a theoretical perspective. To achieve this purpose, we calculate the reward functions achieved by visiting known trajectories and new ones, respectively. The theoretical results demonstrate that the agents achieve more rewards for visiting known trajectories than exploring new ones.

Consider an objective of mutual information between the agent identity $i$ and trajectory $\tau$

$$
\begin{aligned}
I(i;\tau) &= \mathbb{E}_{i,\tau}[\log p(i \mid \tau)] - \mathbb{E}_i[\log p(i)] \\
&\geq \mathbb{E}_{i,\tau}[\log q_\theta(i \mid \tau)] - \mathbb{E}_i[\log p(i)]
\end{aligned}
\tag{8}
$$

where the $p(i \mid \tau)$ is an unknown posterior distribution that is approximated by a variational distribution $q_\theta(i \mid \tau)$ derived by the variational inference approach. $q_\theta(i \mid \tau)$ can be trained via maximum likelihood on $(i, \tau)$-tuples induced by the policy $\pi$ of each agent. To maximize the mutual information, the variational lower bound can be deployed in MARL as an intrinsic reward

$$
\begin{aligned}
r(\tau, i') &= \log q_\theta(i' \mid \tau) - \log p(i') \\
&= \log q_\theta(i' \mid \tau) + \log |A|
\end{aligned}
\tag{9}
$$

where $i' \sim p(i)$, a fixed uniform distribution. Here, $-\log p(i') = \log |A|$ since we have a total number of $|A|$ agents. We assume a perfect variational distribution $q_\theta(i \mid \tau)$ where $\sum_{a=1}^{|A|} q_\theta(i_a \mid \tau) = 1$.

**Reward for known trajectories** The reward function encourages the agents to visit identity-aware trajectories to highlight themselves from others where $q_\theta(i' \mid \tau) \to 1$, thus

$$
r_{\max} = \log 1 + \log |A| = \log |A|.
\tag{10}
$$

**Reward for new trajectories** For unseen trajectories, the value of $q_\theta(i' \mid \tau)$ is unknown. Here, we add a *background* class to the model in order to assign null probability to unseen trajectories. Therefore, the agent receives a penalization for visiting unseen trajectories:

$$
r'_{\text{new}} = \lim_{q_\theta(i' \mid \tau) \to 0} \log q_\theta(i' \mid \tau) + \log |A| = -\infty
\tag{11}
$$

We note that the intrinsic reward enforces the agent to visit known trajectories that contain more identity information, making the agents prone to overfitting and leading to static trajectories.

# B   The CTR implementation on top of MAPPO

In addition to QMIX, MAPPO is another state-of-the-art policy-based MARL algorithm on SMAC. It learns a shared actor network updated towards maximizing expected returns. To integrate with MAPPO, we deploy the CTR model in the actor network and introduce auxiliary gradients derived from the contrastive learning loss to the policy gradients used to update parameters of the actor network. Thus, we can achieve the overall objective that updates the actor network toward maximizing expected returns while minimizing the contrastive learning loss to learn distinguishable trajectory representations

$$
\mathcal{J}_{\text{total}} = \mathcal{J}_{actor} - \alpha \mathcal{L}_N^m,
\tag{12}
$$

where $\mathcal{J}_{actor}$ is the objective of MAPPO to train the actor network. Since CTR only needs to learn distinguishable trajectory representations via training the actor network, we don't need to change the other components of MAPPO. The experimental results of CTR integrated with MAPPO are listed in Table 1, demonstrating the outperformance of our method compared to baselines.

# C   Pseudocode for CTR

The PyTorch-style pseudocode for CTR is given in Algorithm 1.

**Algorithm 1:** PyTorch-style pseudocode for CTR

```
# batch:  collected trajectories
# H: dimension of identity representation
# |A|:  number of agents
identity_representation = linear(H,|A|)
def ctr_loss(batch):
    ctr_out = []
    for t in range(batch.seq_length):
        input_t = concat(batch["obs"][:, t], batch["actions_onehot"][:, t-1])
          # Assemble the inputs
        z_embedding = encoder(input_t)
        c_embedding, hidden_states = autoregressive_model(z_embedding,
         hidden_states)
        ctr_out.append(c_embedding)
    ctr_out = th.stack(ctr_out, dim=1) # Concat over time
    ctr_loss = contrastive_learning_loss(identity_representation, ctr_out,
     |A|) # Calculate the contrastive learning loss.
    return ctr_loss
```

## D   Environmental details and Additional experimental results

This section gives the details of the experimental environments including Pac-Men, SMAC, and SMACv2. We design a grid-world environment Pac-Men to demonstrate the effectiveness of CTR in encouraging multi-agent diversity. In the Pac-Men environment, four agents are initialized at the center of the maze. Each agent has a partial observation and can only observe a $4 \times 4$ grid around them. Each edge room has some randomly initialized dots. During each episode, the agent moves to one of the four edge rooms to eat dots. To improve the difficulty of the task, we set different path lengths for the paths to edge rooms. The path lengths for the downward, left, right, and upward paths are 3, 6, 6, and 10, respectively. Only one path is within the observation scope of the agent, which further challenges the agent's ability to explore the environment. Dots will refresh after all of them are eaten by agents. The reward received by the agent is the number of dots eaten in each step.

SMAC and its upgraded version SMACv2 are the benchmarks for cooperative multi-agent reinforcement learning research. Both of them consist of a variety of fully cooperative tasks that are implemented on top of Blizzard's real-time strategy game StarCraft II to evaluate the effectiveness of various MARL algorithms. SMAC realizes the agent level control using the Machine Learning APIs provided by StarCraft II and DeepMind's PySC2. In each task, there is a combat scenario that involves two armies: one of the armies is controlled by the allied RL agents and the other is controlled by the built-in non-learned game AI. The game ends when all units of any army have died or a pre-defined timestep limit is reached. The goal of the allied agents is to learn a policy that can maximize the win rate of the game. Therefore, the agents need to learn a sequence of actions to collaborate with other allies to defeat the enemy forces. One example of such collaboration involves mastering kiting skills, where all agents form formations according to their armor types, forcing enemy units to pursue and keep enough distance from the enemies to reduce damage. We use the SC2.4.10 version of SC2. Note that the performance is not comparable between versions.

SMACv2 is an upgraded version of SMAC that uses the same APIs as SMAC to control the game units. Unlike SMAC, SMACv2 introduces stochasticity in the StarCraft II environment. Firstly, the initial positions of units are random, challenging the agent to learn to defeat the enemies from different angles. Secondly, the allied agents of each scenario have random unit types instead of pre-specified types. These beneficial changes provide additional challenges for MARL algorithms, as they must learn generalizable and robust policies to improve win rates. The average returns of compared algorithms in Pac-Men, SMAC, and SMACv2 are listed in Table 1.

## E   Evaluations of CTR in scenarios requiring homogeneous behavior

Although behavioral diversity is crucial in the multi-agent environment, agents may sometimes find it beneficial to act similarly in straightforward situations. For instance, allied agents might employ the

Table 1: Average returns of compared algorithms in Pac-Men, SMAC, and SMACv2. $\pm$ denotes the standard deviation over five random seeds.

| Method | Pac-Men | SMAC | | | | | | SMACv2 | | |
|---|---|---|---|---|---|---|---|---|---|---|
| | | 3s5z | 2c_vs_64zg | 7sz | 6h_vs_8z | corridor | 3s5z_vs_3s6z | terran_5_vs_5 | protoss_5_vs_5 | zerg_5_vs_5 |
| QMIX | 0.21±0.04 | 0.72±0.13 | 0.85±0.08 | 0.17±0.02 | 0.23±0.03 | 0.57±0.07 | 0.36±0.12 | 0.68±0.03 | 0.53±0.05 | 0.41±0.04 |
| MAPPO | 0.49±0.03 | 0.81±0.05 | 0.83±0.04 | 0.52±0.06 | 0.53±0.03 | 0.62±0.05 | 0.57±0.08 | 0.52±0.04 | 0.47±0.03 | 0.37±0.03 |
| MAVEN | 0.32±0.06 | 0.51±0.21 | 0.72±0.06 | 0.00±0.00 | 0.17±0.04 | 0.36±0.08 | 0.18±0.15 | 0.58±0.04 | 0.31±0.05 | 0.29±0.03 |
| EOI | 0.41±0.05 | 0.87±0.07 | 0.83±0.02 | 0.37±0.03 | 0.08±0.03 | 0.25±0.11 | 0.42±0.13 | 0.65±0.05 | 0.42±0.03 | 0.47±0.04 |
| QTRAN | 0.28±0.08 | 0.21±0.19 | 0.75±0.05 | 0.00±0.00 | 0.02±0.02 | 0.08±0.07 | 0.02±0.01 | 0.42±0.02 | 0.40±0.04 | 0.25±0.02 |
| SCDS | 0.37±0.05 | 0.76±0.07 | 0.57±0.09 | 0.21±0.03 | 0.03±0.01 | 0.56±0.06 | 0.00±0.00 | 0.52±0.03 | 0.47±0.05 | 0.38±0.04 |
| LIPO | 0.43±0.02 | 0.71±0.03 | 0.76±0.02 | 0.39±0.04 | 0.36±0.06 | 0.27±0.03 | 0.21±0.03 | 0.43±0.02 | 0.46±0.03 | 0.37±0.03 |
| FoX | 0.39±0.03 | 0.74±0.02 | 0.64±0.05 | 0.56±0.03 | 0.45±0.05 | 0.52±0.04 | 0.43±0.04 | 0.54±0.03 | 0.56±0.02 | 0.49±0.02 |
| **CTR+QMIX** | **0.78±0.04** | **0.95±0.03** | 0.87±0.03 | **0.82±0.05** | **0.79±0.03** | **0.82±0.07** | **0.85±0.06** | **0.83±0.03** | **0.75±0.05** | **0.73±0.03** |
| **CTR+MAPPO** | 0.75±0.03 | 0.92±0.04 | **0.89±0.05** | 0.78±0.03 | 0.71±0.04 | 0.78±0.05 | 0.81±0.04 | 0.79±0.04 | 0.69±0.03 | 0.65±0.03 |

Table 2: Performance of our method and QMIX in homogeneous scenarios.

| Method | 8m | 5m_vs_6m | 8m_vs_9m | 10m_vs_11m |
|---|---|---|---|---|
| CTR+QMIX | 0.95±0.03 | 0.93±0.04 | 0.94±0.02 | 0.91± 0.04 |
| QMIX | 0.87±0.03 | 0.65±0.04 | 0.58±0.05 | 0.43±0.04 |

same tactic to simultaneously fire at an enemy to quickly eliminate it. To demonstrate the effectiveness of our method in learning such behaviors, we test it in four homogeneous SMAC scenarios that benefit from the focus fire trick. The results, shown in Table 2, indicate that our method consistently outperforms QMIX across all scenarios. The outperformance of our method demonstrates that it supports, rather than prevents, homogeneous behaviors when these result in higher environmental rewards. Moreover, our method is more efficient to search these optimal cooperative behaviors due to sufficient exploration.

# F   Scalability

In MARL, as the number of agents increases, the state-action space grows exponentially, challenging agents to search optimal collaborative policies. Many algorithms lack efficient exploration and may not scale well with a large number of agents. To demonstrate the scalability of our method, we test it in four SMACv2 scenarios with an increasing number of agents: terran_5_vs_5, terran_10_vs_10, terran_15_vs_15, and terran_20_vs_20. The results are shown in Table 3. The performance of QMIX decreases significantly as the number of agents increases. We believe this is because QMIX suffers from poor exploration of the state-action space. With the help of our method, QMIX achieves essentially better performance and exhibits robust scalability, demonstrating that learning distinguishable trajectory representations for agents to make action decisions leads to efficient exploration.

# G   Training Details and Hyperparameters

Our proposed CTR model consists of an encoder and an autoregressive model. We adopt two stacked resnet blocks with a hidden size of 64 followed by batch normalization for the encoder and a GRU unit for the autoregressive model. In the agent utility network of QMIX, the CTR model encodes a trajectory to a latent representation space, which is then input to the fully connected output layer to calculate the per-agent utilities. Similar to QMIX, in the actor network of MAPPO, the learned trajectory representations are input to a fully connected output layer followed by a softmax function

Table 3: Performance of our method and QMIX in scenarios of SMACv2 with different number of agents

| Method | terran_5_vs_5 | terran_10_vs_10 | terran_15_vs_15 | terran_20_vs_20 |
|---|---|---|---|---|
| CTR+QMIX | 0.85±0.04 | 0.87 ±0.02 | 0.83 ±0.03 | 0.82 ±0.03 |
| QMIX | 0.68±0.03 | 0.39±0.04 | 0.24 ±0.06 | 0.11±0.05 |

to output the distributions over actions. Moreover, we introduce a learnable vector for the identity representation that has the same dimensions as trajectory representations.

We use the same policy network architecture for other baselines as in our method to guarantee a fair comparison. In both SMAC and SMACv2, the target networks are updated via hard updates every 200 episodes. In Pac-Men, the target networks use soft updates at a momentum rate of 0.01. We set the evaluation interval to 10K steps followed by 32 test episodes. We run all methods for 5 million steps. The hyperparameters of CTR and baseline algorithms in Pac-Men, SMAC, and SMACv2 are listed in Table 4. To ensure a fair comparison, we set the same values for common hyperparameters across different methods in each multi-agent environment. Additionally, all methods adopt the parameter-sharing technique to accelerate training speed. For generality, we report the mean and standard deviation of the experimental results tested with five random seeds. We set the replay buffer size to 5K. We implement our method with NumPy and PyTorch. All experiments are performed using NVIDIA GeForce RTX 4090 GPUs.

Table 4: Hyperparameters

|  | Pac-Men | SMAC | SMACv2 |
|---|---|---|---|
| hidden dimension | 64 | 128 | |
| learning rate | 0.0003 | 0.005 | |
| optimizer | | Adam | |
| target update | 0.01(soft) | 200(hard) | |
| batch size | 32 | 64 | |
| $\alpha$ for CTR+QMIX | 0.02 | 0.1 for 3s5z, 2c_vs_64zg, 8m, 5m_vs_6m, 8m_vs_9m, and 10m_vs_11m, 0.02 for 7sz, 6h_vs_8z, corridor, and 3s5z_vs_3s6z | 0.004 |
| $\alpha$ for CTR+MAPPO | 0.1 | 0.1 for 3s5z, 2c_vs_64zg, 8m, 5m_vs_6m, 8m_vs_9m, and 10m_vs_11m, 0.05 for 7sz, 6h_vs_8z, corridor, and 3s5z_vs_3s6z | 0.02 |
| epsilon anneal time | 200,000 | 200,000 for 3s5z, 2c_vs_64zg, 8m, 5m_vs_6m, 8m_vs_9m, and 10m_vs_11m, 500,000 for 7sz, 6h_vs_8z, corridor, and 3s5z_vs_3s6z | 500,000 |

# H   Visualization

We additionally provide visualized t-SNE plots in Figure 8, Figure 9, and Figure 10 to intuitively compare the trajectory representations learned by QMIX, EOI, SCDS, and CTR in the super hard corridor, 6h_vs_8z, and 3s5z_vs_3s6z scenarios. The trajectory representations of different agents learned by CTR ultimately stay away from each other and become distinguishable while those learned by QMIX are mixed. Moreover, although EOI and SCDS also encourage multi-agent diversity, the distinguishability between trajectory representations of different agents learned by them is less pronounced compared to CTR.

To intuitively demonstrate the diverse policies learned by CTR+QMIX, we present some visualization examples of the diverse policies emerging in 6h_vs_8z, corridor, and 3s5z_vs_3s6z from initial to final in Figure 11. Green and red shadows represent agents and enemies, respectively. Green and red arrows represent the moving direction of agents and enemies, respectively. For example, in the 6h_vs_8z scenario, one agent first leaves the team separately to incur the attention of most enemies. The agent keeps kiting the enemies and draws the fire to cover other agents. Then other agents move in different directions and attack the few remaining enemies. If all the agents take similar policies and consistently move toward the enemies, they will be killed immediately. These diverse policies can also be observed in the other two scenarios: corridor and 3s5z_vs_3s6z. These examples suggest that learning diverse policies by distinguishing trajectory representations of different agents finally enables the agents to cooperatively defeat the enemies.

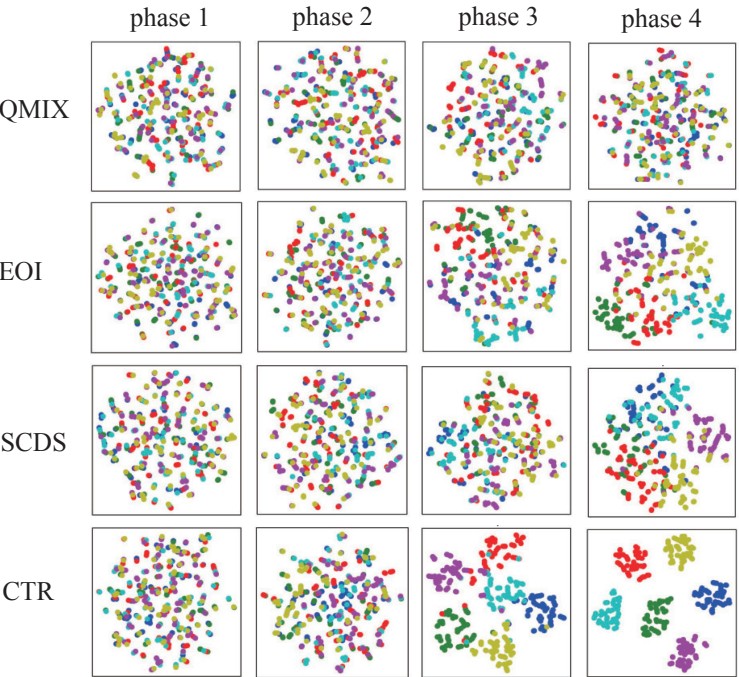

Figure 8: T-SNE plots of trajectory representations of different agents learned by CTR and baselines, respectively, that emerge in the corridor scenario, initial (left) to final (right). Each color represents the trajectory representations of an agent.

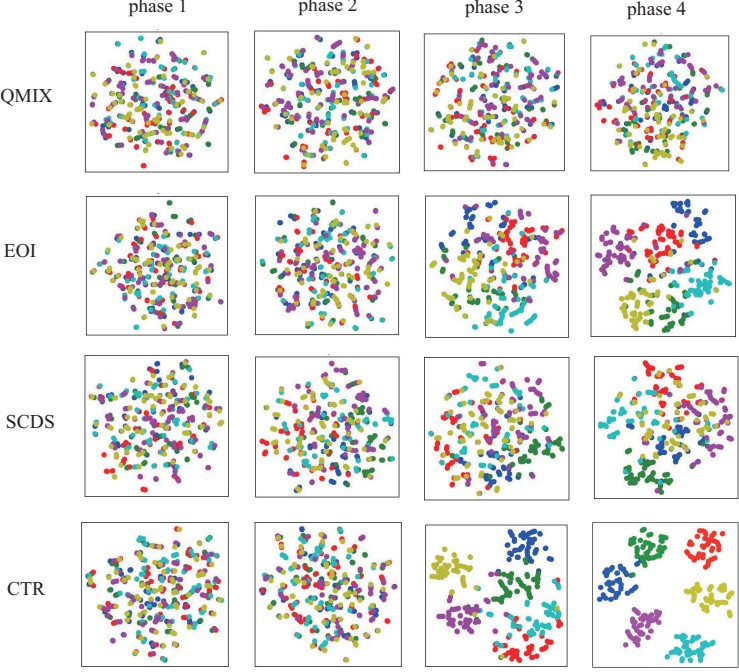

Figure 9: T-SNE plots of trajectory representations of different agents learned by CTR+QMIX and baselines, respectively, that emerge in the 6h_vs_8z scenario, initial (left) to final (right). Each color represents the trajectory representations of an agent.

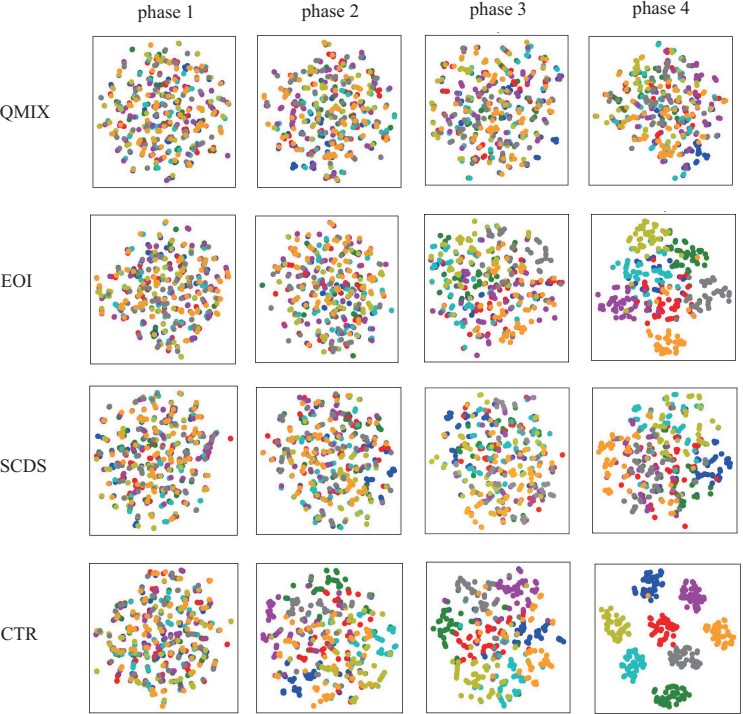

Figure 10: T-SNE plots of trajectory representations of different agents learned by CTR+QMIX and baselines, respectively, that emerge in the 3s5z_vs_3s6z scenario, initial (left) to final (right). Each color represents the trajectory representations of an agent.

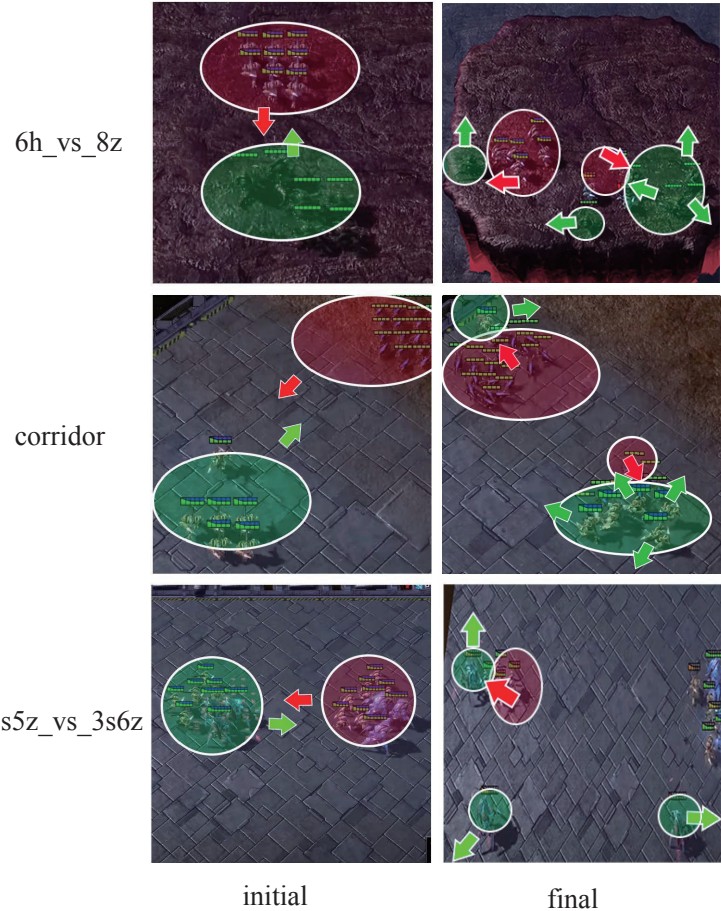

initial

final

Figure 11: Visualization examples of diverse policies emerging in 6h_vs_8z (top), corridor (medium), and 3s5z_vs_3s6z (bottom) from initial (left) to final (right). Green and red shadows represent agents and enemies, respectively. Green and red arrows represent the moving directions of agents and enemies, respectively.

