# OpenReview forum: "Learning Distinguishable Trajectory Representation with Contrastive Loss"
_NeurIPS.cc/2024/Conference — NeurIPS 2024 poster_

### Official Review · Reviewer_u9xg · 2024-07-09

**Soundness:** 3
**Presentation:** 2
**Contribution:** 2
**Rating:** 4
**Confidence:** 3

**Summary:**

This paper presents a contrastive approach for learning diverse policies in multi-agent reinforcement learning (MARL). It maximizes the mutual information between trajectory representations and identity representations, formulating this maximization as an InfoNCE loss function. The methodology is tested across several scenarios, including Pac-Men, SMAC, and SMACv2.

**Strengths:**

- The paper derives and theoretically substantiates the InfoNCE loss function.
- The experimental results demonstrate that the proposed method surpasses the performance of existing baseline methods in tested scenarios.

**Weaknesses:**

This paper has some shortcomings in technical novelty and clarity:

1. **Technical Novelty:** The application of InfoNCE loss to representation learning in this context appears incremental, bearing close resemblance to the previously established CIA method. The distinction between the proposed method and existing techniques is not adequately emphasized, which raises concerns about the paper’s original contribution.
2. **Unclear Claims About Diversity:** The paper does not specify which MARL framework supports the proposed method. Assuming a shared policy network parameter setting similar to QMIX, the rationale behind diverse policies under such a configuration is questionable. The shared parameters typically lead to the same policies, contradicting the stated objective of enhancing policy diversity among multi-agents.
3. **Evaluation of Diversity:** Although diversity is a major goal, the paper lacks specific metrics or evaluations of diversity in the experimental setup. The focus is task performance, but short in directly assessing whether the proposed method actually enhances the multi-agent diversity.

**Minor Issues:**
- The paper incorrectly refers to GRU as an autoregressive model. GRU is a recurrent neural network.
- Figures 3 and 4 are in wrong aspect ratios.

**Questions:**

1. Do agents share the same parameters of the policy network in the proposed method?
2. How is “diversity” defined in the context of this paper? Does it imply that different agents have different policies?
3. How does the proposed CTR method compare to the existing CIA method in terms of performance and policy diversity?

---

> ### Author Rebuttal · Authors · 2024-08-06
>
> Thank you for taking the time to review our paper. We clarify your concerns and problems below:
>
> Weakness 1: Technical Novelty ... contribution.
>
> We discussed the CIA method in the related works in our original paper. The differences between our method and CIA are shown below:
>
> First, our main idea of learning distinguishable trajectory representations to encourage multi-agent diversity is entirely different from the CIA which distinguishes temporal credits of different agents. We learn a trajectory encoder using contrastive learning to output distinguishable trajectory representations, while CIA treats the temporal gradients of agents derived from the TD loss of QMIX as credits and uses the InfoNCE loss to distinguish them to realize identity-aware credit assignment. Note that CIA can only be used in methods based on the value-based value-decomposition framework. However, our method can also be used in policy-based methods such as MAPPO.
>
> Second, CIA and our method are all inspired by the InfoNCE loss proposed in [1]. The use of the InfoNCE loss in the field of MARL can also be found in many other works such as [1], [2], and [3]. CIA directly applies the InfoNCE loss to distinguish temporal credits of different agents. However, most importantly, we do not directly apply the InfoNCE loss but a novel multi-agent contrastive learning loss shown in Equation 4. This is because we need to solve the limitation of applying the InfoNCE loss to learn trajectory representations in multi-agent settings that the smaller dataset size used in our method may induce a larger gap between the true mutual information objective and the contrastive learning lower bound, which hurts the agent performance. So, we improve the InfoNCE loss by actively increasing the number of negative samples in the denominator, which leads to better empirical results.
>
> For the reasons stated above, our method is different from CIA.
>
> Weakness 2: Unclear Claims ... multi-agents.
>
> In our method, all agents share the same policy network parameters. The claim that learning distinguishable trajectory representations for agents, sharing the policy network parameters, to learn diverse behaviors is reasonable. First, since MARL methods such as QMIX typically adopt techniques such as $\epsilon$-greedy or entropy regularizers to introduce randomnesses into action selections during the exploration phase, they may not learn the exact same policies. The agents sharing policy network parameters are likely to achieve similar trajectories as we discussed in the third paragraph of Section 1 in our paper. Agents trained by QMIX typically make action decisions based on its historical trajectories. The intuition behind our method is that although similar historical trajectories among agents can typically lead to similar actions when using the shared policy network to make action decisions, we can instead learn distinguishable trajectory representations from similar historical trajectories for action decision making by training the policy network (CTR is a part of the policy network) towards minimizing the contrastive learning loss. This means that although the input historical trajectories are similar, the shared policy network learns distinguishable trajectory representations, thus leading to diverse policies. Contrastive learning gives us a chance to distinguish trajectories of different agents in a representation space.
>
>
> Weakness 3: Evaluation of Diversity ... diversity.
>
>
> There is currently a lack of commonly used specific metrics to measure diversity among agents. So, we demonstrate the agent diversity as in previous works. To intuitively demonstrate the effectiveness of our method in encouraging multi-agent diversity, we provided some agent's visitation heatmaps, T-SNE plots of trajectory representations, and visualization examples of diverse policies in Figure 2 and Figure 7-11 in our paper.
>
>
> Q1: Do agents share the same parameters of the policy network in the proposed method?
>
> Please refer to Weakness 2.
>
> Q2: How is “diversity” defined in the context of this paper? Does it imply that different agents have different policies?
>
> Multi-agent diversity refers to the diversity among agents, i.e., the differences between policies of different agents. We discussed the “diversity” in the third paragraph of the Section 1.
>
> Q3: How ... policy diversity?
>
> We compare our method with CIA in Pac-Men and the three super hard scenarios of SMAC used in our paper. The hyperparameters and policy network structures are kept consistent across the two methods to ensure a fair comparison. We present the experimental results in Table 4 in the attachment of the global response. Compared to CIA, our method maintains its outperformance across all scenarios. In Pac-Men, CIA fails to discover the top room with the longest path as shown in Figure 1 in the attachment of the global response, thus leading to sub-optimal performance. Our method is more efficient in encouraging multi-agent diversity, which enables agents to go to different rooms to collect dots as demonstrated in Figure 2d in our paper.
>
> We hope to receive your feedback soon and greatly appreciate the time you have taken to review our paper.
>
> [1] Hu, Zican, et al. "Attention-guided contrastive role representations for multi-agent reinforcement learning." arXiv preprint arXiv:2312.04819 (2023).
>
> [2] Zeng, Weihao, et al. "Multi-Agent Transfer Learning via Temporal Contrastive Learning." arXiv preprint arXiv:2406.01377 (2024).
>
> [3] Lo, Yat Long, et al. "Learning Multi-Agent Communication with Contrastive Learning." The Twelfth International Conference on Learning Representations.

---

> > ### Comment · Reviewer_u9xg · 2024-08-11
> >
> > Thanks for your response. I still have some concerns and questions:
> >
> > Weakness 2
> >
> > I am still unclear on how the same policy network can produce "diverse policies," as this is a key motivation of the paper (Section 1 and 3.1).
> > I understand that actions can vary due to the stochastic nature of policies, even if the policies themselves are identical, but the concept of diverse policies as presented remains ambiguous.
> > If we place two agents in the same initial state, would they execute the same policies or different?
> > What specific components of CTR contribute to the diversity of policies during inference?
> >
> > Q2
> >
> > Because all agents share same policy network parameters,  it seems that all policies are the same. The "diversity", defined as "the differences between policies of different agents", seems to be zero.
> >
> > Q3
> >
> > The test win rates reported in Table 4 in the attachment are lower than the original CIA paper in corridor, 6h_vs_8z, and 3s5z_vs_3s6z.
> > Refer to the results in original CIA paper, CIA achieves comparable performance with CTR.
> > Another paper [1] shows experimental results of CIA as baselines method, where CIA's performance is also higher than attachment reported performance in corridor and 3s5z_vs_3s6z.
> > Besides, the results of proposed method CTR in Table 4 in attachment differ from Figure 3 in the main paper, in 6h_vs_8z and 3s5z_vs_3s6z.
> > The authors should provide a detailed explanation of the discrepancy in these results.
> >
> > -----
> >
> > [1] Attention-Guided Contrastive Role Representations for Multi-agent Reinforcement Learning. Hu et al. ICLR 2024

---

> ### Author Response · Authors · 2024-08-12
>
> Thanks for your feedback. We respond to your concerns below:
>
> Weakness 2: I am still unclear on how the same policy network can produce "diverse policies," as this is a key motivation of the paper (Section 1 and 3.1). I understand that actions can vary due to the stochastic nature of policies, even if the policies themselves are identical, but the concept of diverse policies as presented remains ambiguous. If we place two agents in the same initial state, would they execute the same policies or different? What specific components of CTR contribute to the diversity of policies during inference?
>
> The reason why the same policy network can produce "diverse policies," is that agents sharing the same policy network have different inputs (historical trajectory) or learn different trajectory representations. If agents are encouraged to visit diverse trajectories, we may say that they have different policies.
>
> If two agents have the same initial observations, they will not generate exactly the same trajectories due to the existence of the stochasticity in action selections. However, they are likely to visit similar trajectories as observed by many previous works. So, our method uses contrastive learning to train the shared policy network to learn distinguishable trajectory representations from such similar trajectories. By minimizing the contrastive learning loss, the trajectory representations stay close to their corresponding identity representations while being far away from other identity representations, leading to distinguishability among trajectory representations. The distinguishable trajectory representations can then be used to make diverse action decisions, thus leading to diverse trajectories and different policies. We clarified this point in our paper.
>
> Q2: Because all agents share same policy network parameters, it seems that all policies are the same. The "diversity", defined as "the differences between policies of different agents", seems to be zero.
>
> Although agents share the same policy network parameters, they can still have different inputs or learn different representations, which leads to diverse behaviors. These diverse behaviors can typically result in diverse trajectories, thus leading to diverse policies. The concept of diverse policies is relevant to the diverse trajectories that the agents visit.
>
> Q3: The test win rates reported in Table 4 in the attachment are lower than the original CIA paper in corridor, 6h_vs_8z, and 3s5z_vs_3s6z. Refer to the results in original CIA paper, CIA achieves comparable performance with CTR. Another paper [1] shows experimental results of CIA as baselines method, where CIA's performance is also higher than attachment reported performance in corridor and 3s5z_vs_3s6z. Besides, the results of proposed method CTR in Table 4 in attachment differ from Figure 3 in the main paper, in 6h_vs_8z and 3s5z_vs_3s6z. The authors should provide a detailed explanation of the discrepancy in these results.
>
> The differences in the performance of CIA across different papers result from different network structures and different hyperparameters. We use the same network structures and hyperparameters across different baseline methods to ensure a fair comparison. Moreover, the performance results over different SMAC versions are not comparable. The results of our method in 6h_vs_8z and 3s5z_vs_3s6z come from Table 1 in the appendix of our paper, which lists the numerical results of performance. We clarified this in our paper.
>
> We hope to hear from you soon and thank you again for your review.

---

> ### Author Response · Authors · 2024-08-13
>
> We anticipate your response and appreciate your continued attention to our work.

---

> > ### Comment · Reviewer_u9xg · 2024-08-13
> >
> > Several claims in the paper require further clarification. The comparison experiments with CIA are not sufficiently convincing. I have decided to raise my score to 4, but I still maintain a negative stance.

---

### Official Review · Reviewer_mufx · 2024-07-12

**Soundness:** 2
**Presentation:** 3
**Contribution:** 2
**Rating:** 4
**Confidence:** 4

**Summary:**

This paper proposes a novel Contrastive Trajectory Representation (CTR) method based on learning distinguishable trajectory
 representations to encourage multi-agent diversity. It introduces contrastive learning to maximize  the mutual information between the trajectory representations and learnable identity  representations of different agents.

**Strengths:**

- This paper propose a novel strategy to make multi-agent diversity by learning distinguishable trajectory representations.
- CTR maps the trajectory of an agent into a latent trajectory representation space by an encoder and an autoregressive model.
- The proposed distinguishable trajectory representations do not rely on fixed agent identities.

**Weaknesses:**

- The paper lacks a detailed description of the identity representation, only saying that it is a  learnable vector.
- The paper cited the latest developments in reinforcement learning, does not compare their performance [1]. Newer published works in recent 3 years should get included [2].
- The quality of the figures needs to be improved such Fig. 2 and 3. The fonts in figures seem to be strange.
Reference.
[1] Contrastive Identity-Aware Learning for Multi-Agent Value Decomposition, AAAI, 2023.
[2] Learning Multi-Agent Communication with Contrastive Learning, ICLR, 2024.

**Questions:**

- What the differences between the proposed method and these methods[1][2].
- How to get learnable vector $d$ for each agent?
- It encouraged for authors to demonstrate the rewards and loss curves of proposed method.

**Limitations:**

- Lack of implementation details for proposed method. Please make the source codes be available.

---

> ### Author Rebuttal · Authors · 2024-08-06
>
> Thank you for your careful review and for providing us with detailed and helpful feedback. We response to your concerns below:
>
> Weakness 1: The paper lacks a detailed description of the identity representation, only saying that it is a learnable vector.
>
> Previous mutual information-based methods typically use fixed agent identities, e.g., one-hot vectors. However, this may lead to serious overfitting as we discussed in the fourth paragraph of Section 1 in our paper. To solve the problem, we instead use learnable identity representations for agents to represent their identities. The identity representations can be trained by minimizing the contrastive learning loss to linearly classify the trajectory representations of different agents. As a result, the identity representations stay close to their corresponding trajectory representations while being far away from other trajectory representations, leading to distinguishability among trajectory representations. We clarified it in our paper.
>
> Weakness 2: The paper cited the latest developments in reinforcement learning, does not compare their performance [1]. Newer published works in recent 3 years should get included [2].
>
> We conduct a comparison of our method with the CIA method introduced in [1], in Pac-Men and the three super hard scenarios of SMAC. To ensure a fair comparison, we maintain consistent hyperparameters and policy network structures for both methods. The results are presented in Table 4 in the attachment of the global response. Our method consistently outperforms CIA across all scenarios. Specifically, in Pac-Men, CIA learns suboptimal policies and does not identify the top room with the longest path, as illustrated in Figure 1 in the attachment of the global response. Our method enhances multi-agent diversity more effectively, enabling agents to explore different rooms to collect dots, as shown in Figure 2d in our paper.
>
> We included [2] in our related works. The authors in [2] aim to help agents to learn to communicate for better coordination. To achieve this goal, they uses contrastive learning to maximize the mutual information between messages of a given trajectory, which leads the messages from the same state to be more similar to each other than to those from distant states or other trajectories. We also use the contrastive learning loss. However, we use it to learn distinguishable trajectory representations to encourage multi-agent diversity and further improve it to achieve better empirical results.
>
> Weakness 3: The quality of the figures needs to be improved such Fig. 2 and 3. The fonts in figures seem to be strange. Reference.
>
> We resized the fonts in the figures to make them clearer.
>
> Q1: What the differences between the proposed method and these methods[1][2].
>
> CIA proposed in [1] realizes credit-level distinguishability using contrastive learning to distinguish the agents' temporal credits that are represented by the agent's gradient-based attributions derived from the TD loss of QMIX. CACL proposed in [2] learns to communicate using contrastive learning to maximize the mutual information between messages of a given trajectory. However, we learn distinguishable trajectory representations from similar trajectory samples using contrastive learning to encourage multi-agent diversity. Further, CIA and CACL directly use the original InfoNCE loss proposed in [3]. We improve the original InfoNCE loss in multi-agent settings, as shown in Equation 4, by increasing the number of negatives, leading to better empirical results.
>
> Q2: How to get learnable vector $d$ for each agent?
>
> At the beginning of the training process, we randomly initialize alearnable identity representation $d^a \in \mathbb{R}^H$ for each agent by building a LearnableDictionary class that wraps a torch.nn.Parameter function in Pytorch to generate learnable tensors.
>
> Q3: It encouraged for authors to demonstrate the rewards and loss curves of proposed method.
>
> We provide episode returns (the sum of rewards achieved by agents during an episode) and loss curves in Figure 2 and Figure 3 in the attachment of the global response, respectively.
>
> Limitations: Lack of implementation details for proposed method. Please make the source codes be available.
>
> We provide the PyTorch-style pseudocode for our method and training details in Section C and Section G, respectively. Our source codes, included as supplemental materials, were uploaded when we submitted this paper.
>
> [1] Liu, Shunyu, et al. "Contrastive identity-aware learning for multi-agent value decomposition." AAAI, 2023.
>
> [2] Lo, Yat Long, et al. "Learning Multi-Agent Communication with Contrastive Learning." ICLR, 2024.
>
> [3] Oord, Aaron van den, Yazhe Li, and Oriol Vinyals. "Representation learning with contrastive predictive coding." arXiv preprint arXiv:1807.03748 (2018).

---

> > ### Comment · Reviewer_mufx · 2024-08-10
> >
> > Thanks for your responses, according to the responses and other reviewers' comments, I will keep my score.

---

### Official Review · Reviewer_GQbC · 2024-07-12

**Soundness:** 3
**Presentation:** 3
**Contribution:** 2
**Rating:** 6
**Confidence:** 3

**Summary:**

The paper proposes using a contrastive trajectory representation to improve diversity and exploration in decentralized multi-agent reinforcement learning. Experimental evaluations show the positive impact in a small-scale environment and improved performance in various SMAC scenarios.

**Strengths:**

The authors provide a varied and insightful evaluation, showing the positive impact of their approach, which aims to improve MARL diversity and explorative capabilities, which constitutes a relevant problem in MARL by integrating constrastive learning.

**Weaknesses:**

- The authors should state more clearly their use of parameter sharing in a decentralized learning scenario to avoid the confusion of only a single policy with shared parameters being learned.
- The proposed method and its intentions could be explained in greater detail (e.g., the consequences and effect of Eq. (4) or the difference between the contrastive learning loss and the InfoNCE loss in Eq. (2)).
- The proposed architecture and setting are very similar to AERIAL (Phan et al., 2023), so they should be compared. Overall, the use of mutual information to foster diverse policy or trajectory representations has been used in various approaches and does not seem novel (which generally is no issue to me but should be discussed).
- Conclusion and Limitations fall pretty short and could be extended, e.g., by potential issues of increasingly complex learning architectures.

Minor Comments:

- 2.1 could also be a paragraph, given there is no further subsection in 2.
- in 2.1: Does $U$ refer to the joint action or the joint action space?

**Questions:**

If agent identities are randomly chosen, what effect does the distance or mutual information between the trajectory representation and this random vector have?

How does your approach improve limited exploration?

**Limitations:**

Limitations are briefly discussed but can be extended (see above).

---

> ### Author Rebuttal · Authors · 2024-08-06
>
> We greatly value your expertise and the effort you put into reviewing our paper. Here are the responses to your concerns:
>
> Weakness 1:The .. being learned.
>
> In our paper, we discuss a decentralized learning scenario where agents share the same policy network parameters but learn different decentralized policies. Thank you for your thoughtful suggestions. We clarified it in our paper.
>
> Weakness 2: The ... in Eq. (2)).
>
> We discuss the motivation, the consequences, and the effect of using Equation 4 in Section 3.3. We developed the multi-agent contrastive learning loss shown in Equation 4 because the small size of dataset $C$, equal to the number of agents, results in a wider gap between the true mutual information objective and the contrastive learning lower bound, potentially hurting performance. So we propose the multi-agent contrastive learning loss, which actively increases the number of negative samples from $O(|\mathcal{C}|)$ to $O\left(|\mathcal{C}|^2\right)$, thus leading to better empirical results. We clarified this further in our paper. Contrastive learning loss is a broad category of loss functions that includes the InfoNCE loss. We typically use contrastive learning loss to demonstrate the employment of contrastive learning.
>
> Weakness 3: The ... be discussed).
>
> The only similarity between AERIAL proposed in [1] and our method is that we all use the hidden states or trajectory representations output by the RNNs. However, their applications are entirely different. AERIAL processes the hidden states using a simplified transformer in order to automatically learn the latent dependencies over hidden states through self-attention, and then feed the output of the transformer to the mixing network instead of the true state like QMIX. Our method has a totally different objective that encourages multi-agent diversity through learning distinguishable trajectory representations via contrastive learning.
>
> The idea of learning distinguishable trajectory representations to encourage multi-agent diversity is novel and has not been proposed yet. Moreover, although many previous methods employ the mutual information objective to encourage exploration to improve the cooperation among agents, however, they typically resort to the variational lower bound to solve the mutual information objective, which may lead to serious overfitting as we discussed in the fourth paragraph of Section 1 and Section A in the appendix. We solve this limitation of the mutual information-based methods by using contrastive learning to learn to distinguish trajectory representations. The empirical results demonstrate that our method can lead to better performance.
>
> Weakness 4: Conclusion ... architectures.
>
> We add some limitations as follows: As our method needs to contrast all agents' trajectories, this necessitates that the training process should be centralized so that the trajectories of all agents can be collected. Thus, our method cannot be applied to fully decentralized MARL methods. Moreover, in contrastive learning, by collecting many negative samples, the model is challenged to distinguish the positive pair from a larger pool of negatives. This helps the model learn more robust and discriminative features. Although we developed multi-agent contrastive learning loss to increase the number of negative samples, however, the number of negative samples is still limited if the total number of agents in multi-agent environments is very small. For our future work, we may develop efficient methods to augment existing trajectory samples to increase the number of samples.
>
> We add some conclusions as follows: Our simple yet effective method demonstrates the importance of representation learning in promoting efficient exploration. Better representation learning can typically encourage the learning optimal policies. Despite the emergence of multi-agent diversity, we also note the need for homogeneous behaviors. Although our method would not impede the learning of homogeneous behaviors that can lead to more environmental rewards, how to control diversity automatically can be an interesting direction for our future work.
>
> $U$ is a set of agents’ actions as we discussed in Section 2.1.
>
> Q1: If agent ... have?
>
> The choice of agent identities would not affect the trajectory representation learning. In our paper, we use the learnable identity representations to represent the agent identities, which are randomly initialized are the beginning of the training process. From the InfoNCE loss shown in Equation 3, we note that by minimizing the InfoNCE loss, the identity representations stay close to their corresponding trajectory representations while being far away from other trajectory representations. Thus, the distance between the trajectory representation and the identity representation is irrelevant to the choice of agent identities.
>
>
> Q2: How does your approach improve limited exploration?
>
> Different from the previous mutual information-based methods that learn the mutual dependence between trajectories and fixed agent identities, e.g., fixed one-hot vectors, to encourage multi-agent diversity, which may lead to serious overfitting, our method learns distinguishable trajectory representations by minimizing the contrastive learning loss. The learned trajectory representations do not depend on fixed agent identities, thus leading to more efficient exploration. Moreover, from the T-SNE plots shown in Figure 8-10, we note that previous mutual information-based methods such as SCDS and EOI do not learn distinguishable trajectory representations, thereby resulting in inefficient exploration. Compared to previous methods, our method successfully learns distinguishable trajectory representations with large distances.
>
> We hope to hear from you soon and thank you again for your review.
>
> [1] Phan, Thomy, et al. "Attention-based recurrence for multi-agent reinforcement learning under stochastic partial observability." ICML. PMLR, 2023.

---

> > ### Comment · Reviewer_GQbC · 2024-08-12
> >
> > Thank you for your extensive response. My concerns are mostly addressed, and I hope some of the detailed explanations are integrated into the final version.
> >
> > However, I still agree with reviewer u9xg that a single shared policy network cannot be referred to as "diverse". As I understand your responses, the preprocessing of histories causes diverse representations, which improves exploratory capabilities. I therefore suggest adding further clarifications to the paper.
> >
> > Overall, the paper still provides a valuable contribution, and I will maintain my original score.

---

> ### Author Response · Authors · 2024-08-12
>
> Thank you for your feedback. The concept of "diverse" discussed in our paper implies diverse trajectories among different agents. Agents sharing the same policy network parameters may lead to diverse trajectories if the policy network has different inputs (historical trajectories) or learns different trajectory representations. As the agents sharing the same policy network parameters tend to achieve similar trajectories, directly using such similar trajectories as inputs of the shared policy network may lead to similar behaviors. Our method thus learns distinguishable trajectory representations using contrastive learning for action decision making to encourage diverse behaviors among different agents, leading to the visitations of diverse trajectories. If agents visit diverse trajectories, we believe they are learning diverse policies. We clarified this point in our paper.

---

### Official Review · Reviewer_ncsG · 2024-07-21

**Soundness:** 3
**Presentation:** 3
**Contribution:** 3
**Rating:** 7
**Confidence:** 2

**Summary:**

This paper introduces a novel approach to learning in multi-agent reinforcement learning (MARL) environments by focusing on distinguishable trajectory representations to encourage agent diversity. The proposed method, termed Contrastive Trajectory Representation (CTR), leverages a contrastive learning loss to effectively differentiate between agents' trajectory representations without necessitating fixed agent identities. This approach is designed to combat the common issue in MARL where agents converge to similar behaviors, thus limiting the overall system's adaptability and efficiency.

The paper provides a comprehensive evaluation of CTR through a series of experiments conducted in both grid-world environments like Pac-Men and more complex settings such as the StarCraft Multi-Agent Challenge (SMAC). The results demonstrate that CTR significantly outperforms existing state-of-the-art methods by facilitating more robust and diverse agent behaviors, leading to improved exploration and performance across various scenarios. This is achieved by mapping trajectories onto a contrastive representation hypersphere, which encourages more efficient exploration and prevents the overfitting associated with mutual information-based methods. The study underscores the effectiveness of CTR in promoting diverse and adaptive strategies in MARL settings, offering a scalable solution that enhances both the learning efficiency and strategic capabilities of multi-agent systems.

**Strengths:**

The paper introduces an innovative approach within the field of multi-agent reinforcement learning (MARL) by focusing on the creation of distinguishable trajectory representations using contrastive learning. This method diverges from traditional reliance on mutual information maximization between agents' identities and trajectories, presenting a significant shift in how agent diversity is cultivated in MARL environments. The originality of this work is evident as it creatively combines the principles of contrastive learning, commonly employed in single-agent domains or supervised learning tasks, with the complexities of MARL, thereby addressing the challenge of agent homogenization without fixed identity assignments.

The quality of the research is high, evidenced by rigorous experimental design and thorough validation across several benchmarks, including grid-world scenarios and the StarCraft Multi-Agent Challenge (SMAC). The experiments are well-structured, with clear comparisons to baseline models and detailed discussions of the results. Furthermore, the authors have provided a robust statistical analysis to back their claims, reinforcing the reliability of their findings. The use of well-recognized MARL environments for testing also supports the methodological rigor of the study.

The paper is well-written, with a clear exposition of the concepts and methodology. The authors have successfully communicated complex ideas in an accessible manner, making the paper understandable to both experts in the field and readers with a more general background in machine learning.

**Weaknesses:**

W1: Dependency on Hyperparameters: The performance of the CTR model, as with many learning models, appears to be  dependent on the fine-tuning of hyperparameters, particularly the weight of the contrastive loss component. This dependency can introduce challenges in scenarios where the optimal hyperparameter settings are not apparent or vary significantly between environments. The paper could enhance its contribution by providing a more detailed analysis or guidelines on how to select or adapt these parameters effectively across different settings.

W2 Scalability to Larger Agent Pools: The experiments are somewhat limited in scale, primarily focusing on scenarios with a small to moderate number of agents. The scalability of the approach to environments with large numbers of agents remains untested. It would be beneficial to investigate how the method performs as the number of agents increases significantly, which is a common challenge in real-world applications of MARL.

W3 Handling Non-Stationarity: In MARL environments, the non-stationarity issue arises when the policies of other agents change, which can affect the learning and stability of a given agent’s policy. The paper perhaps does not address how the proposed method copes with the non-stationarity of the environment, which is crucial for ensuring robustness and reliability in dynamically changing scenarios.

**Questions:**

Q1: I'm wondering whether environments like football games can get benefit as you mentioned. Is it possible to make such experiments?
[1]TiKick: Towards Playing Multi-agent Football Full Games from Single-agent Demonstrations

Q2: Could you elaborate on the mathematical or theoretical foundations that justify the use of contrastive learning specifically in MARL environments? How does this approach theoretically ensure improved agent diversity compared to traditional methods?

Q3: How does the model perform over extended periods of interaction? Is there evidence of long-term stability in the agents' behaviors, or do they exhibit significant variance over time?

**Limitations:**

Probably no obvious limitations

---

> ### Author Rebuttal · Authors · 2024-08-06
>
> We appreciate your time and the valuable insights you provided during the review process. We clarify your concerns and problems below:
>
> W1: Dependency ... different settings.
>
> The values for the weight of the contrastive loss in different scenarios are listed in Table 4 in our paper. To investigate the effect of the weight of the contrastive loss component in different scenarios, we use different weight values and test them in the easy scenario 3s5z and the super hard scenario corridor. The results are shown in Table 1 in the attachment of the global response. We note that smaller values for the weight of the contrastive loss can typically lead to better empirical results in the super hard scenario, while larger values are welcome in the easy scenario. This may be attributed to the environmental rewards achieved by agents in the super hard scenario being less than those achieved in the easy scenario. Therefore, less intrinsic rewards are desirable in the super hard scenario. Moreover, our method is not very sensitive to the values of the weight. Sub-optimal weights do not result in a significant performance drop even in the super hard scenario.
>
> W2: Scalability ... of MARL.
>
> We evaluated the scalability of our method in different scenarios of SMACv2 with an increasing number of agents from 5 to 20. Our method scales well as the number of agents increases. We further test the scalability of our method in a large-scale multi-agent benchmark IMP-MARL [1]. Specifically, we use the environment uncorrelated k-out-of-n system from IMP-MARL with the number of agents varying from 10 to 100. We present normalised relative rewards achieved by our method and QMIX, respectively, in Table 2 in the attachment of the global response. Our method substantially achieves better performance than QMIX and scales well with a varying number of agents. A larger number of agents results in more negative samples that are desired to improve the discrimination of contrastive representation learning.
>
> W3: Handling ... changing scenarios.
>
> In our method, we use an autoregressive model to encode the trajectory representations for action decision-making, which compresses the historical trajectory information. In multi-agent settings, the historical trajectory information can typically alleviate the non-stationary issue. Moreover, the InfoNCE loss learns trajectory representations by considering the trajectory representations of all agents, thus leading to more stable and robust policy learning.
>
> Q1: I'm ... Demonstrations
>
> Yes. We further test our method on two challenging Google Research Football (GRF) offensive scenarios academy\_3\_vs\_1\_with\_keeper and academy\_counter\_attack\_hard. We present the results in Table 3 in the attachment of the global response. We note that our method achieves significant performance improvement over QMIX, demonstrating the effectiveness of our method in encouraging efficient exploration.
>
> Q2: Could ... methods?
>
> The learning rule of the MARL method theoretically guarantees the learning of optimal policies. Our method does not break the learning rule of the integrated MARL method since its implementation is within the decentralized policy network. For example, our method does not break the monotonicity constraint of QMIX imposed on the mixing network. This constraint ensures that the gradient of $Q_{tot}$ with respect to any individual $Q_a$ is non-negative: $\frac{\partial Q_{\text{tot}}}{\partial Q_i} \geq 0$. This guarantees that improvements in an agent’s policy that increase its own Q-value will not decrease the total Q-value. The monotonicity constraint is enforced by architecting the mixing network so that it uses non-negative weights. However, our method only introduces an auxiliary gradient derived by the contrastive learning loss to the individual agent utility network to learn distinguishable trajectory representations, that would not affect the value decomposition controlled by the mixing network.
>
> The diversity measures the distance between the trajectories of different agents. We may associate such distances with mutual dependence from the perspective of information theory. We first model a density ratio as in CPC[2]: $\frac{p(c_{t}^a | d^a)}{p(c_{t}^a)}$ that preserves the mutual information between the trajectory representation $c_{t}^a$ and identity representation $d^a$. Then, we let the similarity $f\left(c_t, d\right)= \exp \left({c_t^a}^T d^a \right) \in \mathbb{R}$ be proportional to the ratio: $ f\left(c_t, d\right) \propto \frac{p(c_{t}^a | d^a)}{p(c_{t}^a)}$, which can be achieved by minimizing the InfoNCE loss shown in Equation 3 as theoretically proved by CPC. This demonstrates that the InfoNCE loss models $\frac{p(c_{t}^a | d^a)}{p(c_{t}^a)}$ instead of $p(c_{t}^a | d^a)$. Previous methods typically aim to maximize the probability $p(c_{t}^a | d^a)$ to achieve the maximum of the mutual information. We notice that $\frac{p(c_{t}^a | d^a)}{p(c_{t}^a)} > p(c_{t}^a | d^a)$. Thus, our method can be more efficient in distinguishing the trajectory representations of different agents.
>
> Q3: How ... over time?
>
> We typically set a large time-step limit for each episode in test environments. For example, we run 100 steps per episode in Pac-Men. We provide visitation heatmaps of our method in Pac-Men in Figure 2d. We also present the visualization examples of policies learned by our method from initial to final in three super hard scenarios of SMAC in Figure 11. These results demonstrate that our method can continuously encourage agents to explore and exhibit efficient diverse policies over time.
>
> We await your further suggestions.
>
> [1]Leroy, Pascal, et al. "IMP-MARL: a suite of environments for large-scale infrastructure management planning via MARL." Advances in Neural Information Processing Systems 36 (2024).
>
> [2]Oord, Aaron van den, Yazhe Li, and Oriol Vinyals. "Representation learning with contrastive predictive coding." arXiv preprint arXiv:1807.03748 (2018).

---

> > ### Comment · Reviewer_ncsG · 2024-08-11
> > **Response to the authors**
> >
> > Thank you for the authors. My questions are majorly solved, and I will keep my rate.

---

### Official Review · Reviewer_Urvi · 2024-07-31

**Soundness:** 3
**Presentation:** 4
**Contribution:** 3
**Rating:** 7
**Confidence:** 3

**Summary:**

The authors propose a method to maximize mutual information between trajectory representations of different agents in the multi-agent reinforcement learning setting. Rather than comparing policies on a state-by-state basis, they instead learn a representation of the entire trajectory using sequence models and contrastive learning. They evaluate their method in a range of challenging multi-agent RL games and show that their method accelerates learning and achieves higher final rewards than baselines.

**Strengths:**

The paper is well-written and it is easy to understand the main concepts. The experiments show strong advantages of the method in multiple settings, providing strong evidence that the method is making a significant difference. In addition to the strong comparisons to baselines, the authors also provide insightful visualizations and ablate over their design choices.

**Weaknesses:**

In theory, this method can be applied to other algorithms besides QMIX. It could be used to learn distinguishable skills in unsupervised skill learning, or in continuous control settings as well. There is no need to limit the thinking and future work to this setting

**Questions:**

N/A

**Limitations:**

Yes

---

> ### Author Rebuttal · Authors · 2024-08-06
>
> We thank you for taking the time to review. We agree with your comments. Our method can be integrated with a variety of MARL methods based on the CTDE framework including value-based and policy-based methods. Since our method simply incorporates the trajectory encoder with the decentralized policy network to learn distinguishable trajectory representations, it is irrelevant to other network structures of MARL methods. Moreover, our method can be extended to learn diverse skills in the field of unsupervised skill learning. Prior works focus on maximizing the mutual information between states and latent variables to learn diverse skills, which may lead to overfitting as we discussed in our paper. Similar to our work, learning distinguishable representations for different states may be more efficient in learning diverse skills. We reserve this for our future work.

---

> > ### Comment · Reviewer_Urvi · 2024-08-12
> > **Rebuttal Response**
> >
> > I thank the authors for their response

---

### Author Rebuttal · Authors · 2024-08-07

We greatly appreciate the time you have taken to review our paper. The PDF attachment presents the Tables and Figures referenced in the responses. We hope to receive your feedback soon so that we can further improve our paper.

---

### Decision · Program_Chairs · 2024-09-25

**Decision:**

Accept (poster)

**Comment:**

The paper proposes a method for improving behavior diversity in MARL by having the agents learn different trajectory representations using a contrastive loss-based approach CTR while using a shared policy network. The experimental results show that CTR significantly improves the performance of the underlying MARL algorithm. While contrastive learning has been widely for inducing trajectory representations, the specific approach in this paper is novel. Given the performance boost it affords to MARL, the metareviewer recommends this paper for acceptance.